# Beyond Thresholds: Multi-Modal Graph-Based Learning for Predictive Scoring in Preclinical Alzheimer's Disease

Nupur Thakur*, Riti Paul*, Yuxiang Zhou[†], Oana Dumitrascu[†], Baoxin Li*
*Arizona State University, Tempe, Arizona, USA
[†]Mayo Clinic, Phoenix, Arizona, USA

*Abstract*—The regression task of predicting preclinical Alzheimer's disease (AD) risk using imaging and other data is essential for evaluating the varying risk levels across individuals. Existing data-driven methods for identifying preclinical AD primarily focus on binary classification, which relies on thresholds derived from specific imaging datasets. This reliance on dataset-specific thresholds can hinder the generalization capability of such methods across datasets, limit the integration of other types of data, and restrict the assessment of varying risk levels among individuals. In this paper, we propose a novel, multi-modal, semi-supervised regression framework to predict amyloid positivity scores by integrating brain MR imaging and non-imaging data, such as genetic information and cognitive assessments. We introduce an unsupervised, imaging data-driven module for prototype label generation, for the 'regression by classification' learning strategy. We employ graph neural networks to model the complex relationships within diverse non-imaging data. Our learning algorithm makes optimal use of the labeled and unlabeled data, maximizing the utility of limited labeled information, while eliminating the need for rigid thresholds. Through extensive evaluation on the ADNI dataset, which includes real-world patient data, our framework demonstrates effectiveness, suggesting that it could offer a more adaptable, precise tool for preclinical AD assessment and a step forward in the application of computer vision and deep learning to neurodegenerative disease detection.

*Index Terms*—Graph Neural Network, MR Imaging, Multi-modal Learning, Preclinical Alzheimer's Disease, Semi-supervised Learning

## I. Introduction

Computer vision has advanced neurodegenerative disease research [1]–[4]. Alzheimer's disease (AD), which primarily affects older adults, progresses through three stages: preclinical (asymptomatic), Mild Cognitive Impairment (MCI), and Alzheimer's Dementia. While deep learning has been widely applied to detect MCI and dementia stages [1], [5], preclinical AD detection remains underexplored. This is of particular significance, as research indicates that early identification and intervention can potentially slow down or even prevent the progression of AD.

Individuals in the preclinical AD stage are usually labeled as 'cognitively normal' with minimal detectable neurodegeneration, posing key detection challenges. First, subtle brain changes often go unnoticed by the unaided human eye. Second, early AD markers like Amyloid beta (A$\beta$) deposition [6], [7] require costly, invasive tests (e.g., PET scans, lumbar

punctures), which are not usually done for cognitively normal patients. Instead, magnetic resonance imaging (MRI), a less invasive, more affordable option, is commonly used as a surrogate. Third, amyloid positivity differences are subtle since preclinical AD patients are labeled as cognitively normal. Fig. 1 shows SUVR score distributions for preclinical AD and cognitively normal patients from ADNI data. Capturing these nuances is essential for accurate risk assessment.

Existing MRI-based methods for detecting preclinical AD [8], [9] typically treat it as a binary classification problem, ignoring individual variability in risk levels. For example, [9] uses predefined thresholds to assign labels, making results sensitive to small data changes. Predicting a continuous risk score instead, offers clinicians more actionable insights

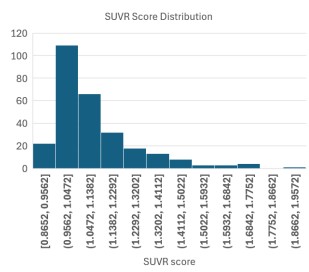

Fig. 1. Target score (SUVR) distribution from ADNI dataset. These scores fall within a narrow range and exhibit very subtle variations.

and greater flexibility for medical decision-making, advantages missing in classification approaches.

To address these challenges, we propose a novel semi-supervised regression approach for preclinical AD risk scoring using brain MRI and non-imaging data like demographics and genetics. The risk score approximates the standardized uptake value ratio (SUVR) from amyloid-PET (see Related Work for elaboration). We introduce a novel unsupervised prototype generation technique for regression by classification, enabling the use of both labeled and unlabeled data, unlike prior methods [10] limited to labeled data. Our model employs graph neural networks to capture relationships among non-imaging features and leverages semi-supervised learning to handle limited SUVR annotations. Evaluated on the ADNI dataset, our method effectively distinguishes preclinical AD patients from cognitively normal, producing precise risk scores with valuable insights.

*To the best of our knowledge, this is the first deep learning-based risk-scoring approach, specifically for preclinical AD.*

Our key contributions to address the important challenges are (i) **Regression-based predictive scoring approach** for preclinical AD accounting for individual risk differences, making it applicable to a wider range of clinical encounters and giving the medical professionals flexibility to adjust the parameters; (ii) **Data-driven prototype label generation** for regression by classification learning to recognize subtle differences between the risk scores of cognitively normal individuals and those in the preclinical stage of AD; (iii) **Graph-based modeling for non-imaging data** to capture the relations within this diverse data; (iv) **Learning from multimodal data** to overcome the weakness of imaging biomarkers for preclinical AD.

## II. RELATED WORK

Early detection of Alzheimer's Disease (AD), particularly differentiating mild cognitive impairment (MCI) from advanced stages, is crucial for personalized treatment and early clinical trial enrollment [11]. To enhance diagnostic precision and support early intervention, research has increasingly focused on robust classification methodologies to identify MCI and more advanced AD stages. [12] proposed a four-class SVM classifier to categorize AD, stable and progressive MCI, and healthy individuals. Additional works, such as [2], [13]–[19], have furthered this progress by exploring advanced architectures, including RNN, Vision Transformers (ViT), and hybrid models like convolutional neural networks (CNN) with transformers, to enhance diagnostic capabilities.

In contrast to MCI or AD, where there are noticeable or significant brain structural changes, detecting preclinical AD is a relatively more complex task due to the lack of noticeable brain structural changes. Detecting preclinical AD before MCI develops is crucial, as it offers an opportunity for early intervention.

Due to the challenging nature of the problem statement, research in this area is scarce. [8] was among the first to explore preclinical AD detection in the OASIS-3 dataset, proposing two attention-based networks, demonstrating the effectiveness of MR imaging for classification. [20] highlighted the role of non-invasive multimodal data (EEG, APOE4, demographics, neuropsychology, MRI) in predicting amyloid using Random Forest, logistic regression, and SVM. [9] leveraged amyloid positivity as an early preclinical AD indicator, proposing a HexaGAN-based generative framework for data imputation and amyloid prediction in cognitively normal patients, though they relied on a predefined threshold that may vary across cohorts. Most existing studies focus on classification, which requires a well-defined dataset.

Amyloid beta ($A\beta$) deposition is widely recognized as a key biomarker for Alzheimer's Disease (AD) due to its association with the disease's early pathological changes, especially during the preclinical stages [6], [21]. Research indicates that amyloid plaques, formed by $A\beta$ aggregation, begin to develop in asymptomatic individuals, which correlates with progressive neurodegeneration and cognitive decline [22]. Studies using positron emission tomography (PET) imaging and cerebrospinal fluid (CSF) biomarkers have demonstrated that elevated $A\beta$ levels are detectable in the preclinical phase, offering a potential window for preventive interventions [23]. Given its strong correlation with preclinical AD, we use SUVR scores as risk indicators.

## III. PROPOSED METHOD

The key objective of this work is to build a learning-based risk-scoring approach for preclinical AD that approximates the SUVR score from amyloid-PET, using MR imaging and other non-imaging data. We begin by defining the problem in Section III-A, followed by detailed descriptions of the proposed model in subsequent subsections.

### A. Problem Statement

Let $I$ be an MRI volume where $I \in \mathbb{R}^{C \times A \times B}$ and $C, A, B$ are the number of slices, height, and width of the MRI volume respectively, and let $X$ be the non-imaging data where $X \in \mathbb{R}^Q$ such that $Q$ is the number of non-imaging attributes. Given a labeled dataset $D_l = \{I_i^l, X_i^l, S_i\}$, $i \in [1, N]$ and an unlabeled dataset $D_{ul} = \{I_i^{ul}, X_i^{ul}\}, i \in [1, M]$, the task is to learn a predictive model that can be used to produce a preclinical AD risk score $S$, as close to the SUVR score as possible, for a test sample.

### B. Overview of the Proposed Approach

Our approach integrates MRI and diverse non-imaging data to predict preclinical AD risk using a 3D CNN, graph neural network, and advanced attention mechanism. Fig. 2 shows the prediction model (top row) and the data-driven prototype generation process (bottom row) that defines the initial prototypes.

We use a regression by classification technique, where each sample is associated with a prototype plus an offset to produce the final risk score (top row of Fig. 2). Prototypes represent natural clusters learned in an unsupervised manner from imaging data (bottom row of Fig. 2), each with a unique ID (a pseudo label for the prototype) and a mean risk score computed from labeled samples in that cluster. Details of prototype learning are in Section III-C.

In the Risk Prediction module (Fig. 2, top), a 3D U-Net encoder extracts features from the MRI volume, which are combined with non-imaging data graph features to predict the prototype and offset scores. These yield the final preclinical AD risk score (Eq. 4), where higher scores indicate greater preclinical AD risk. Details are in Section III-D.

### C. Data-driven Prototype Generation

To improve precision and capture subtle risk score differences, we adopt a regression by classification method inspired by multi-bin loss [10], which has proven effective in other regression tasks [24]. This is crucial as the risk score differences between cognitively normal and preclinical AD patients are often subtle (refer Fig. 1). Directly dividing the risk score (target) space into bins, as in [10], [24], is unsuitable here because the resulting decision boundaries may not align with clinical criteria, risking misclassification. Such discrepancies are particularly concerning because they could lead to cases

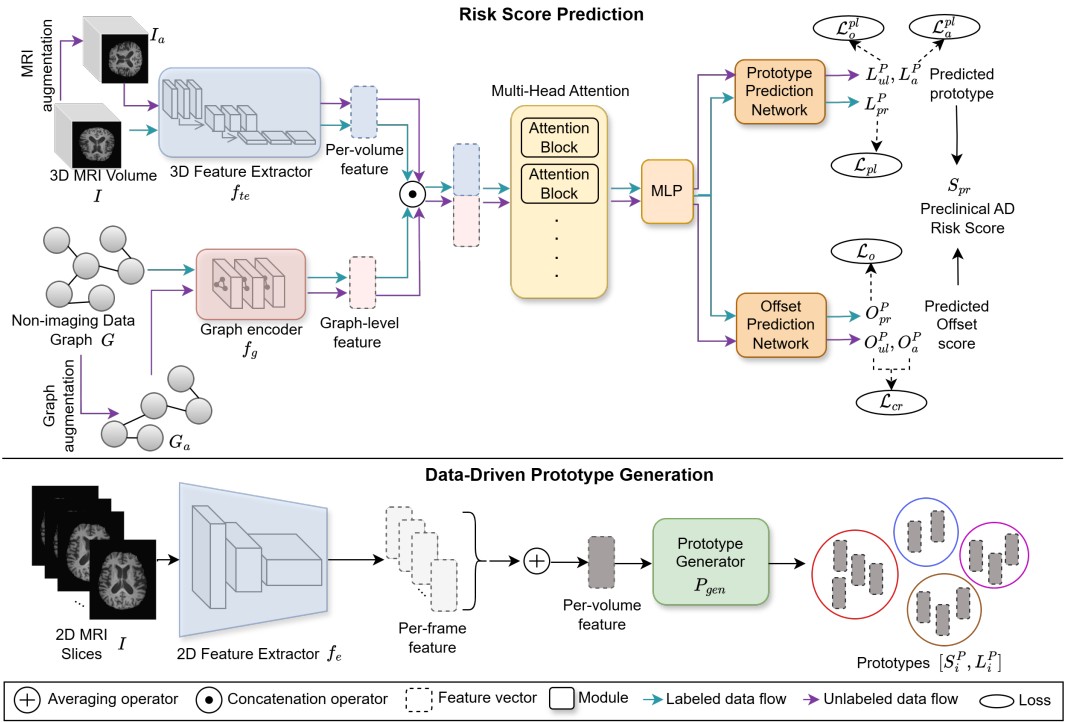

Fig. 2. Overview of the proposed approach. In the Risk Score Prediction module (top), a 3D encoder and a graph encoder extract features from MRI and non-imaging data, respectively. An attention mechanism combines them to predict a prototype label and offset, yielding the final risk score. In prototype generation (bottom), 2D MRI slice features are used to create prototypes, each with a pseudo label and associated risk score.

of preclinical AD going undetected. Moreover, such methods cannot leverage unlabeled data, which is often abundant than labeled data.

In this work, we employ *a data-driven approach to generate prototypes, based on natural groupings of the MR imaging data* in a feature domain. The prototypes, each encoding an ID and a SUVR score, provide a label space for training a regression by classification model. By defining the label space using features rather than manual partitioning (bins), our method reduces the risk of misclassification.

As shown in Fig. 2 (bottom), prototype generation begins by extracting features from each 2D MRI slice of the 3D volume $I$ using a pre-trained 2D feature extractor $f_e$. We use 2D extractors as they are readily available, avoiding the need for pre-training 3D or multi-modal models. A global feature for $I$ is then computed by averaging features across all $C$ slices. The extracted features are fed into the Prototype Generator $P_{gen}$, which uses K-means to assign prototype labels $L^P$ by clustering the data into $k$ clusters. This unsupervised process uses both labeled ($D_l$) and unlabeled ($D_{ul}$) data. Each prototype is defined by its cluster and mean SUVR score, computed from labeled samples. The offset score $O$ is then calculated as the deviation from the corresponding prototype mean $S^P$.

Given an MRI volume $I_i$, the prototype label $L_i^P$, offset score $O_i$ is given by,

$$O_i = S_i - S_j^P, \quad j = L_i^P \qquad (1)$$

$$S_j^P = \frac{1}{\text{num}(K_j)} \sum_{h=1}^{\text{num}(K_j)} S_{jh}, \quad j \in [1, k] \qquad (2)$$

$$L_i^P = P_{gen}\left( \frac{1}{C} \sum_{j=1}^{C} f_e(I_{ij}) \right), \qquad (3)$$

where $\text{num}(K_j)$ is the number of samples in the cluster represented by prototype $K_j$ and $S_j^P$ is the mean risk score of the prototype $K_j$.

### D. Risk Score Prediction

We first process imaging and non-imaging data separately, then fuse them to predict the preclinical AD risk score. A 3D U-Net [25] encoder, $f_{te}$, extracts a $d_1$-dimensional feature from the 3D MRI volume.

For non-imaging data $X$, we construct a fully connected graph $G$, where nodes represent attributes and edges model their relationships, unlike prior work [9], which simply concatenates these attributes. From Fig. 3, we can observe that cognitive assessment scores ADAS13 and MMSE are highly correlated with the age of a patient. Our graph representation of $X$ helps capture and learn from such relations. The attribute value is the initial node embedding. While we explore learnable edge weights, we use equal weights in experiments due to the small attribute set and comparable results (see Section IV-C). However, learnable weights may offer advantages with an increasing number of attributes. A graph encoder module $f_g$ consisting of graph convolutional and pooling layers processes

these initial embeddings. A graph-level $d_2$-dimensional feature is obtained for the non-imaging data $X$ by averaging the processed node embeddings.

We concatenate the imaging and non-imaging features and pass them through a multi-head attention block, allowing the model to focus on the most relevant information. The attention output is then processed by a multi-layer perceptron to predict the prototype label $L_{pr}^P$ and offset score $O_{pr}$. The final predicted risk score $S_{pr}$ is computed as,

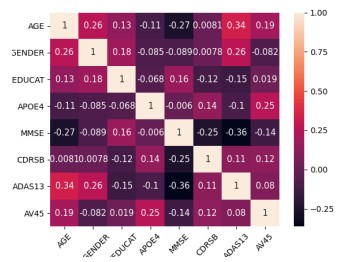

Fig. 3. Correlation between non-imaging attributes & risk scores (ADNI).

$$S_{pr} = S_j^P + O_{pr} \ , \quad j = L_{pr}^P \tag{4}$$

We use multi-class hinge loss to train the Prototype Prediction branch, encouraging larger separation between prototypes for finer score distinctions. The offset prediction branch is trained with mean squared error (MSE) loss.

For labeled data $D_l$, target prototype labels $L_l^P$ are obtained from the Prototype Generation module, and target offset scores $O_l$ are computed from ground-truth risk scores $S_l$ using Eq. 3. The total training loss $\mathcal{L}_l$ for labeled data is,

$$\mathcal{L}_l = \lambda_1 \ \mathcal{L}_{pl} + \lambda_2 \mathcal{L}_o \ , \tag{5}$$

$$\mathcal{L}_{pl} = \sum_{j \neq L_l^P} \max(0, \delta - y_{L_l^P} + y_j) \tag{6}$$

$$\mathcal{L}_o = \frac{1}{N} \sum_{i=1}^{N} (O_l^i - O_{pr}^i)^2 \tag{7}$$

where $\mathcal{L}_{pl}$ is the multi-class hinge loss, $\mathcal{L}_o$ is the MSE loss for the offset scores, $y$ is the output of the Prototype Prediction branch, $\delta = 1$ is the margin hyperparameter and $\lambda_1 = 1, \lambda_2 = 2$ are the weights for individual loss terms.

In the unlabeled dataset $D_{ul}$, target risk scores are not available. To use this data, we apply a semi-supervised technique, consistency regularization [26], encouraging the model to align features from different augmentations of the same input, thus learning meaningful patterns without supervision. For each sample in $D_{ul}$, let $I_{ul}$ represent a 3D MRI volume and $G_{ul}$ a graph of the non-imaging data $X_{ul}$. We create augmented versions, $I_a$ and $G_a$, by applying transformations to these inputs. To generate $I_a$, we apply a random horizontal or vertical flip to the entire MRI volume $I_{ul}$. For $G_a$, we use node dropout, where we randomly remove nodes and their edges from $G_{ul}$ with a probability of 0.5.

Similar to labeled data, we use prototype labels $L_{ul}^P$ obtained from the Prototype Generation module to guide learning on the unlabeled data. We apply a multi-class hinge loss on both the original and augmented inputs using these prototype

labels. To align the offset score predictions for the original and augmented data, we use consistency regularization loss that brings the offset scores closer together. Altogether, the training loss for the unlabeled data, $\mathcal{L}_{ul}$, is computed as,

$$\mathcal{L}_{ul} = \lambda_3 \ \mathcal{L}_o^{pl} + \lambda_4 \ \mathcal{L}_a^{pl} + \lambda_5 \mathcal{L}_{cr} \tag{8}$$

$$\mathcal{L}_o^{pl} = \sum_{j \neq L_{ul}^P} \max(0, \delta - y_{L_{ul}^P}^{ul} + y_j^{ul}) \tag{9}$$

$$\mathcal{L}_a^{pl} = \sum_{j \neq L_{ul}^P} \max(0, \delta - y_{L^P{ul}}^{a} + y_j^{a}) \tag{10}$$

$$\mathcal{L}_{cr} = \frac{1}{M} \sum_{i=1}^{M} (O_{ul}^i - O_a^i)^2 \tag{11}$$

where $\mathcal{L}_o^{pl}$ and $\mathcal{L}_a^{pl}$ represent the multi-class hinge loss calculated on the original unlabeled data and the augmented data, respectively, $\mathcal{L}_{cr}$ is the consistency regularization loss, $\delta = 1$ is the margin hyperparameter for the hinge loss. The weights for each loss term are $\lambda_3 = 1$, $\lambda_4 = 1$, and $\lambda_5 = 2$. We assign a higher weight to the offset score prediction loss, as it captures small deviations from the mean score, which is essential for predicting precise risk scores. The overall training loss $\mathcal{L}$ for our framework is,

$$\mathcal{L} = \Lambda_1 \mathcal{L}_l + \Lambda_2 \mathcal{L}_{ul} \tag{12}$$

where $\Lambda_1, \Lambda_2$ are the weights for the labeled and unlabeled training loss, respectively. We set $\Lambda_1 = \Lambda_2 = 1$ for our experiments.

## IV. EXPERIMENT RESULTS

### A. Experimental Settings

**Dataset details:** For evaluation, we use the Alzheimer's Disease Neuroimaging Initiative (ADNI) [27] dataset, a large, real-world data widely used in Alzheimer's research. We select participants from the ADNI-1, ADNI-2, ADNI-3, and ADNI-GO cohorts who have T1-weighted MRI scans from their screening visit, resulting in a total of 697 cognitively normal (CN) individuals. The AV45 SUVR scores from the ADNIMERGE file are used as the target risk scores.

We use seven non-imaging attributes (Table I) relevant to identifying preclinical AD, covering genetics, cognition, and demographics. Demographic features include age, gender, and years of education. Genetic information is represented by the number of APOE $\epsilon$4 alleles. Cognitive assessments include Mini-Mental State Examination (MMSE), Clinical Dementia Rating-Sum of Boxes (CDR-SB), and Alzheimer's Disease Assessment Scale, the 13-item version (ADAS13).

Out of the 697 samples, 279 have AV45 SUVR scores. This is our labeled data $D_l$ ($N = 279$), in which 92 patients are identified as preclinical AD and 187 as CN, based on a threshold of 1.11 for SUVR scores [9]. However, we do not use this threshold or these labels in any part of our proposed approach or evaluation. The remaining 418 samples ($M = 418$) form our unlabeled data set $D_{ul}$.

**Data preprocessing:** We use T1-weighted MRI scans preprocessed using the following steps: align each scan to the

TABLE I
NON-IMAGING DATA ATTRIBUTES AND SUVR SCORES. FOR CONTINUOUS
VALUES, WE REPORT THE MEAN AND STANDARD DEVIATION. THE
PERCENT OF MISSING VALUES IS CALCULATED USING TOTAL OF 697. M/F
DENOTES MALE OR FEMALE.

| Attribute | Value | Missing (%) |
|---|---|---|
| Age | 71.89±6.73 | 0 |
| Gender | M/F | 0 |
| Education (years) | 16.46±2.49 | 0 |
| APOE $\epsilon4$ | 0, 1, 2 | 11.9 |
| MMSE | 28.94±1.33 | 3.15 |
| CDR-SB | 0.08±0.38 | 3.29 |
| ADAS13 | 8.86±4.60 | 4.01 |
| SUVR | 1.10±0.17 | 70.83 |

standard MNI template using FSL FLIRT [28]; perform skull stripping using FSL BET [29]; normalize the intensity values to the range of 0 to 1. After preprocessing, each MRI volume is of size $182 \times 218 \times 182$.

For the non-imaging data, there are a total of 7 attributes as listed in Table I. The gender attribute is encoded as a categorical feature: 0 represents female, and 1 represents male. Excluding attributes with missing values is not ideal. So, we use the K-Nearest Neighbors (KNN) data imputation method (from the Scikit-Learn library) with 3 neighbors to fill in these missing values. After data imputation, we normalized all values from 0 to 1. To represent the non-imaging data $X$ as a graph $G$, each of the seven attributes is treated as a node with its normalized value as the initial embedding. All nodes are fully connected, resulting in 49 edges.

**Our framework:** The architecture of our approach is shown below: each layer or module of the approach is denoted in the form of $m(d)$, where $m$ is the module/layer and $d$ is the output dimension for $m$.

```
f_te: E_u(16)→E_u(32)→B_u(64)→ FC(64)
E_u: CK(8)→BN(8)→CK(16)→BN(16)
     →PK(16)
B_u: CK(8)→BN(8)→CK(16)→ BN(16)

f_g: FC(16)→GC(64)→NR(64)→ PL(64)

Feature Fusion: MA(128)→FC(64)
Prototype Prediction Network: FC(k)
Offset Score Prediction Network: FC(1)
```

$E_u$: Convolutional block in $f_{te}$, $B_u$: Bottleneck block in $f_{te}$, CK: Convolution layer with a kernel size of $3 \times 3 \times 3$, BN: Batch normalization layer, PK: Max-pooling with kernel $2 \times 2 \times 2$, GC: Graph convolutional layer, NR: Instance normalization layer, PL: Graph Max Pooling layer, MA: Multi-head attention, FC: Fully-connected layer.

We use ImageNet pre-trained ResNet-50 [30] as 2D feature extractor in the Data-Driven Prototype Generation. $k = 5$ for Prototype Generator and the multi-head attention has 4 heads.

**Evaluation metrics:** We evaluate regression performance using Root Mean Squared Error (RMSE) and Mean Absolute Error (MAE), measuring differences between predicted and target risk scores. Reported results are the mean and standard deviation over 5-fold cross-validation.

### B. Baselines for Comparison

Since existing methods [8], [9] only predict labels, not risk scores, direct comparison is not possible. So, we designed simple baseline models that predict risk scores and trained them using Mean Squared Error (MSE) loss.

**Baseline 1:** This baseline uses only the 7-dimensional non-imaging data, as existing methods [9] use it. The input is passed through a fully connected layer with ReLU activation to predict risk scores.

**Baseline 2:** This baseline uses only MR imaging data. It employs a 3D encoder with three $E_u$ blocks and a $B_u$ block, followed by a fully connected layer to predict risk scores.

**Baseline 3:** This baseline encodes image and non-image data using the same networks as the previous baselines. The resulting $d$-dimensional features are concatenated and passed through a fully connected layer to predict the risk score.

### C. Results and Discussion

TABLE II
EXPERIMENT RESULTS ON ADNI DATASET. THE MEAN AND STANDARD
DEVIATION ACROSS 5 FOLDS ARE REPORTED.

| Method | RMSE (↓) | MAE (↓) |
|---|---|---|
| Baseline 1 | 0.675±0.039 | 0.627±0.041 |
| Baseline 2 | 0.384±0.102 | 0.289±0.073 |
| Baseline 3 | 0.359±0.044 | 0.288±0.028 |
| Ours | **0.169±0.013** | **0.121±0.006** |

**ADNI Results:** Table II shows the results on the ADNI dataset. Our framework significantly outperforms all baselines, cutting errors by atleast half across both metrics. It also achieves the lowest standard deviation across the 5 folds, demonstrating both effectiveness and stability.

For further analysis, we show a residual plot of our test data in Fig. 4, illustrat-

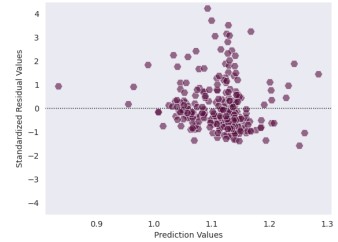

Fig. 4. Residual Plot. Data points cluster near low residual values (close to the horizontal line), indicating predictions closely matching true risk scores.

ing the relationship between predicted risk scores and standardized residuals (difference between true and predicted scores). Most predictions cluster near lower residuals, indicating strong alignment with actual risk scores.

TABLE III
ABLATION STUDY FOR OUR METHOD. ✓ DENOTES THE COMPONENT IS
INCLUDED, AND ✗ REPRESENTS THE EXCLUDED COMPONENTS.

| $I$ | $X$ | $D_l$ | $D_{ul}$ | RMSE ($\downarrow$) | MAE ($\downarrow$) |
|-----|-----|-------|----------|---------------------|--------------------|
| ✗ | ✓ | ✓ | ✓ | 0.171±0.017 | 0.126±0.014 |
| ✓ | ✗ | ✓ | ✓ | 0.190±0.016 | 0.140±0.014 |
| ✓ | ✓ | ✓ | ✗ | 0.170±0.016 | 0.133±0.014 |
| ✓ | ✓ | ✓ | ✓ | **0.169±0.013** | **0.121±0.006** |

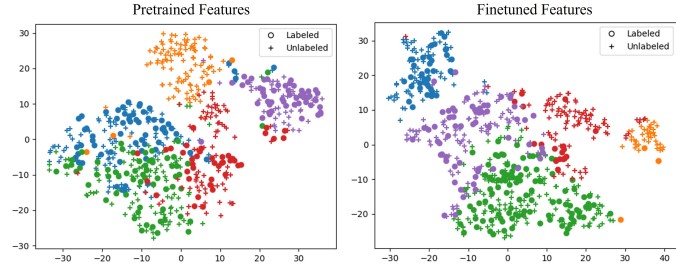

Fig. 5. t-SNE plots for pre-trained and finetuned features. Different colors represent different prototype clusters.

**Ablation Study:** To evaluate the impact of each component, we performed an ablation study (Table III). Removing non-imaging data leads to a significant rise in both error metrics, while excluding MR imaging has a smaller effect, which is intuitive as non-imaging features are stronger preclinical AD indicators. Still, MRI helps reduce errors, highlighting its value when non-imaging data is missing (see Table I). Excluding unlabeled data notably increases MAE, highlighting its contribution in the semi-supervised setting. The best performance is achieved by combining all components. Notably, even when any single component is removed, the results still outperform all baseline methods in Table II.

**2D Feature Extractor for Prototype Generation:** In Data-Driven Prototype Generation, we use ResNet-50 pre-trained on ImageNet to extract features of MR images. To better align with MRI characteristics, we also finetuned the 2D feature extractor on MRI data. To finetune the ResNet-50, we update only the last two convolutional blocks and the fully connected layer, keeping the rest frozen. The model is finetuned for binary classification (preclinical AD vs. cognitively normal) using a stricter SUVR threshold to improve class separation: scores $< 0.95$ labeled as cognitively normal (class 0) and scores $> 1.5$ as preclinical AD (class 1). Data augmentation included random horizontal and vertical flips on the entire MRI volume, and the network was trained for 15 epochs using cross-entropy loss.

TABLE IV
EVALUATION METRICS WHEN DIFFERENT FEATURES ARE USED FOR
PROTOTYPE GENERATION. SSE STANDS FOR SUM OF SQUARED ERRORS
OF THE PROTOTYPE GENERATOR.

| Features | RMSE | MAE | SSE |
|----------|------|-----|-----|
| Pre-trained | 0.169±0.013 | 0.121±0.006 | 1166.74±4.72 |
| Finetuned | 0.165±0.017 | 0.121±0.013 | 1477.18±866.05 |

Table IV shows results using both pre-trained and finetuned features for the Prototype Generator. We report the sum of squared errors (SSE) for Prototype Generator and RMSE and MAE for the proposed method. Finetuned features slightly reduce error metrics but increase variance, especially in SSE, due to using different finetuned models in each validation fold.

Fig. 5 shows t-SNE visualizations (with $k = 5$) for one cross-validation fold, using pre-trained features (left) and finetuned features (right). Both yield well-defined clusters, though some are more compact with finetuned features. However, due to the added computational cost and sensitivity to data distribution shifts without significant error reduction, we choose to use the pre-trained feature extractor in our approach.

**Relationship between non-imaging attributes:** We model non-imaging data $X$ as a fully connected graph $G$ with equal edge weights. To evaluate connection importance, we make the edge weights learnable, yielding an RMSE of 0.165±0.017, comparable to fixed weights (Table II). As shown in Fig. 6, learned weights highlight stronger links (e.g., CDR-SB and MMSE, Age and MMSE) and weaker ones (e.g., MMSE and APOE4), though all remain above 0.98, confirming predictive relevance. We use fixed weights, but note that learnable weights may help when adding more attributes.

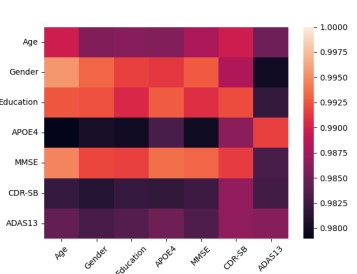

Fig. 6. Visualization of learned edge weights in the non-imaging data graph.

**Varying $k$ for Prototype Generation:** We use K-means clustering in the Prototype Generator, where $k$ must be specified. As shown in Fig. 7, SSE decreases with larger $k$, with a notable drop at $k = 7$. However, higher $k$ can lead to empty clusters due to limited data. To balance SSE, stability, and feature separation, we select $k = 5$ empirically.

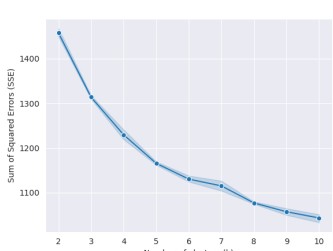

Fig. 7. Plot of varying $k$ in Prototype Generator versus the Sum of Squared Errors.

## V. CONCLUSION AND FUTURE WORK

Our multi-modal, semi-supervised regression framework provides a flexible and accurate approach for assessing preclinical AD risk without relying on fixed classification thresholds. By integrating imaging, demographic, genetic, and cognitive

data using graph neural networks, it enables nuanced risk prediction and addresses limitations in current detection methods.

We validate our approach using a publicly available dataset based on real patients and plan to collaborate with clinical partners for evaluation on real-world cohorts. Since non-imaging data may not always be fully available in practice, we used K-Nearest Neighbor imputation to handle missing entries. Exploring more advanced imputation methods is a promising direction for future work. Our prototype generation module, based on imaging data and 2D feature extractors, effectively guided risk prediction. Including non-imaging data and 3D feature extractors could further improve the prototype quality. Another future direction is applying explainability techniques like Shapley values to enhance model interpretability. Overall, we hope our multi-modal, semi-supervised regression framework serves as foundation for future research in this field.

**Acknowledgement**: R. Paul and B. Li were supported in part by the following grants: NIH RF1AG073424, P30AG072980, 1T32AG082658-01A1, and ADHS Grant #CTR057001. Any opinions expressed in this material are those of the authors and do not necessarily reflect the views of any sponsors.

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
