# OpenReview forum: "Beyond Thresholds: Multi-Modal Graph-Based Learning for Predictive Scoring in Preclinical Alzheimer's Disease"
_IEEE.org/EMBS/BHI/2025/Conference — BHI 2025_

### Official Review · Reviewer_Emtf · 2025-06-30
**Good paper on preclinical AD diagnosis**

**Confidence:** 3
**Clarity Of Writing:** good
**Clinical Significance:** great
**Methodological Novelty:** great
**Overall Rating:** 7
**Final Rating:** 7

**Experiments And Results:**

good

**Questions For The Authors:**

1. Are the prototypes generated based on the entire dataset, including both labelled and unlabelled data?
2. Why use the multi-class hinge loss, as each sample is assigned to only one cluster?
3. How do RMSE and MAE change with respect to the number of clusters?
4. Since a majority of samples (70.83%) miss the SUVR, how does the performance change when excluding the unlabelled samples?

**Strengths:**

1. This paper proposes a multimodal framework to extract features from 3D MRI and graph-like tabular features.
2. This paper proposes a data-driven prototype generation module based on K-means, which defines clusters by their labels and an offset score.
3. The authors generalize the framework to the unlabeled data, offering the advantage of using large-scale data to train the model.

**Summary Of The Paper:**

This paper proposes to tackle the subtle differences in biomarker positivity for preclinical AD diagnosis. The authors propose a risk scoring framework. By transforming the classification to regression, the method has strong flexibility to make diagnosis.

**Weaknesses:**

1. More details about the prototype prediction could be included (detailed in the Questiosn section).
2. The authors used customly implemented methods as baselines as "existing methods only predict labels". However, it would be straightforward to adapt existing methods to do the regression by simply changing the loss function and output head.
3. Implications of the generated prototypes can be further explored. For example, what is the average risk score and the distribution of each prototype cluster? Does any brain region show significant prominance in a cluster?

---

### Official Review · Reviewer_7kRb · 2025-07-08
**Innovative Approach - Clear articulation**

**Confidence:** 5
**Clarity Of Writing:** good
**Clinical Significance:** good
**Methodological Novelty:** good
**Overall Rating:** 7

**Experiments And Results:**

good

**Questions For The Authors:**

I believe these reviews can make the paper better,
1. Can you specify how existing methods are specific threshholds in their training or test dataset and how your model is different from that setup?
2. Please adjust the Figure sizes proportionately with the write up so that they are clearly visible.
3. Adjust Figure 2 to the same page with the overview of Proposed approach where it is referred. Also try to sync the flow of your explanation with Figure 2 or vice versa.
4. What is the necessity for processing imaging and non-imaging data separately?
5. Can you specify what is the need for preprocessing MRI Scan data for Experimental Study?

**Strengths:**

1. Innovative approach for solving the problem.
2. Clear and concised Articulation of the problem statement and methodology.
3. Comparison with the state of the art.
4. Multi-modal data-driven model Implementation

**Summary Of The Paper:**

The paper introduces a semi-supervised, multi-modal regression framework to predict preclinical Alzheimer’s disease risk using MRI and non-imaging data. Unlike prior threshold-based classifiers, it estimates continuous SUVR scores via prototype-based learning and graph neural networks. Tested on the ADNI dataset, the method outperforms baselines and offers a more flexible, accurate tool for early AD risk assessment.

**Weaknesses:**

1. Figures not adjusted according to the text flow.
2. It would be great get some experimental study results on real life case studies, even within a small cohort.
3. Some rational missing about the specific implementation of the proposed framework.

---

### Official Review · Reviewer_831K · 2025-07-18
**Beyond Thresholds: Multi-Modal Graph-Based Learning for Predictive Scoring in Preclinical Alzheimer's Disease**

**Confidence:** 5
**Clarity Of Writing:** great
**Clinical Significance:** good
**Methodological Novelty:** good
**Overall Rating:** 7

**Experiments And Results:**

good

**Questions For The Authors:**

Why SSE metric was specifically used for Prototype Generation Block (Table IV) ? Why was it was not used for the other experiments e.g. baselines in Table II?

**Strengths:**

The  thought behind approaching this problem as a regression problem to identify the subjects at risk of AD is interesting. The method has been tested with the ADNI dataset which is of good size. The label generation using the Prototype Generation method is interesting alongwith the offset score prediction from the score prediction network. Predicting the pseudo label in the final network acts as an auxiliary task and helps the network learn the features better. The ablation table is helpful in understanding the importance of each componen of the network.

**Summary Of The Paper:**

This paper proposes a semi-supervised regression framework leveraging MRI imaging data and non-imaging data such as demographics and genetics for predicting SUVR score which is a clinical predictor of Alzheimer's Disease. The method is a two step method. In the first step unsupervised clustering method K-means to cluster the features of the MRI volumes into 5 clusters (5 is chosen empirically). The features of the MRI volumes are extacted using a framework similar to multi-instance learning i.e. features from the individual slices are learnt and then combined using averaging. The clustering method assigns a pseudo label and associatd risk score to the volume. In the next step, both the pseudo label and associated risk scores are used for training the classification+regression method. The method is shown to outperform existing baselines by adapting those baselines since direct comparison was not possible. The features are learnt via convolutional neural network and the non-imaging featuers are learnt via the graph neural networks. Most of the methods for Alzheimer's Disease focus on the classification task. But formulating the task as a regression task has its advantage of identifying the subjects at risk of AD who are still considered to have normal cognitive ability.

**Weaknesses:**

From the image it looks like the score prediction framework uses 3D MRI volumes though the prototype generation framework uses 2D slices. Can the authors share the insight behind this?
I think the differences between the pretrained features and fine-tuned features in Table IV are minute in terms of RMSE and MAE and it seems the third metric SSE has been specially considered only for this table. The SSE metric absent in the Table II.

---

### Official Review · Reviewer_XrB7 · 2025-07-21
**Technical aspects may be improved**

**Confidence:** 4
**Clarity Of Writing:** good
**Clinical Significance:** good
**Methodological Novelty:** good
**Overall Rating:** 6
**Final Rating:** 7

**Experiments And Results:**

good

**Questions For The Authors:**

Can the graph network be pruned to discard weights of lower magnitude

Although a risk score does not require a threshold per se, wouldn't it still benefit to incorporate it?

**Strengths:**

Good combination of various modalities of data

Use of graph neural networks is interesting

**Summary Of The Paper:**

A multi-modal, semi-supervised regression framework is developed for predicting preclinical AD risk without requiring a threshold for classification.

**Weaknesses:**

No mention of the interpretability of the proposed approach

The use of graph neural networks should be leveraged to enhance interpretability